# Vision-Language Model Selection and Reuse for Downstream Adaptation

**Hao-Zhe Tan** [1 2]  **Zhi Zhou** [1]  **Yu-Feng Li** [1 3]  **Lan-Zhe Guo** [1 2]

## Abstract

Pre-trained Vision-Language Models (VLMs) are becoming increasingly popular across various visual tasks, and several open-sourced VLM variants have been released. However, selecting the best-performing pre-trained VLM for a specific downstream task is challenging since no single VLM can achieve promising performance on all downstream tasks, and evaluating all available VLMs is impossible due to time and data limitations. To address this problem, this paper proposes a novel paradigm to select and reuse VLM for downstream tasks, called **M**odel **L**abel **L**earning (**MLL**). The proposal contains three key modules: *model labeling*, which assigns labels to each VLM to describe their specialty and utility; *model selection*, which matches the requirements of the target task with model labels; and *model reuse*, which applies selected VLMs to the target task in an ensemble manner. The proposal is highly computationally efficient and growable since the model labeling process is completed target task independent and the ability could grow with the number of candidate VLMs. We also introduce a new benchmark for evaluating VLM selection methods, including 49 VLMs and 17 target task datasets. Experimental results clearly demonstrate the effectiveness of the proposed method for selecting and reusing VLMs.

## 1. Introduction

Vision-Language Models (VLMs), such as CLIP (Radford et al., 2021), ALIGN (Jia et al., 2021), which are pre-trained on large-scale image-text datasets, have recently attracted significant attention due to their remarkable zero-shot pre-

diction capabilities on visual tasks. However, though VLM shows impressive general ability, as highlighted in Radford et al. (2021), VLMs often fall short of supervised expert models in many downstream tasks. To address this limitation, numerous studies (Dosovitskiy et al., 2021; Yu et al., 2022; Fang et al., 2023) enhanced the zero-shot performance of VLMs by studying model architectures, pre-training datasets, and training/fine-tuning methods. This effort has led to the development of many open-source pre-trained VLMs with diverse structures and parameters, contributing to VLM model hubs like open-clip (Ilharco et al., 2021), which currently hosts more than 100 pre-trained VLMs.

With the increasing availability of open-source VLMs, selecting and reusing a pre-trained model has become a practical alternative to training one from scratch, which demands extensive training data and computing resources (Li et al., 2021), struggling with catastrophic forgetting during the incremental refinement of the trained model in open environments (Guo et al., 2025). Consequently, a critical problem naturally arises: how to select a VLM to reuse for specific downstream tasks. Although we can directly utilize the best-performing model on a universal dataset such as ImageNet, previous work (Fang et al., 2022) has shown that the performance of VLMs can vary greatly depending on dataset domain. For example, we evaluate the performance of various pre-trained VLMs in the open-clip library across several downstream tasks (1(a)) and within different classes of a specific task (1(b)). Figure 1(a) reveals that each VLM demonstrates distinct strengths in zero-shot visual tasks, with no single model outperforming all others across every task. Interestingly, models that perform worse on general tasks can sometimes surpass stronger models in specific downstream tasks. Furthermore, even in the same task, different VLMs exhibit varying levels of performance across specific classes, as illustrated in Figure 1(b).

Therefore, it is important to design VLM selection methods, and it would be better if we could achieve more fine-grained selection, i.e., select different VLMs to handle different classes. The direct way to select a model is to evaluate all candidate models' performance on the target task. However, it is unrealistic due to time and computational resource limitations. Additionally, previous works on model selection (Tran et al., 2019; You et al., 2021) primarily focus on single-modal models, making them unsuitable for VLM

[1]National Key Laboratory for Novel Software Technology, Nanjing University, China [2]School of Intelligence Science and Technology, Nanjing University, China [3]School of Artificial Intelligence, Nanjing University, China. Correspondence to: Yu-Feng Li <liyf@nju.edu.cn>, Lan-Zhe Guo <guolz@nju.edu.cn>.

*Proceedings of the $42^{nd}$ International Conference on Machine Learning*, Vancouver, Canada. PMLR 267, 2025. Copyright 2025 by the author(s).

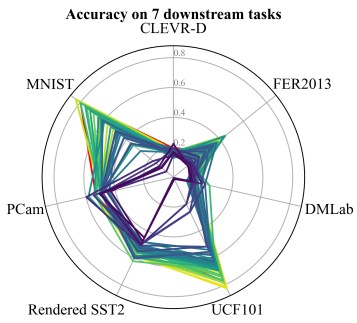
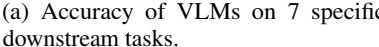
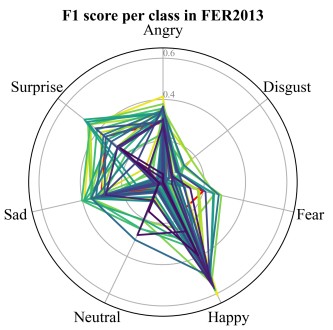

(a) Accuracy of VLMs on 7 specific downstream tasks.

(b) F1 score of VLMs per class in FER2013.

*Figure 1.* The spider charts measure 49 models' capabilities across 7 downstream tasks and classes within a task, showing that the best-performing models vary across downstream tasks and classes, highlighting the importance of model selection for VLM. The evaluated 49 models align with those in the model hub, as discussed in Section 5.1.

selection since they only handle either image or text output and cannot incorporate data from the other modality. Zohar et al. (2023) is the first study to focus on VLM selection, proposing to evaluate VLM performance using textual information. However, the selection strategy heavily depends on the models' ground-truth performance on large-scale datasets, such as ImageNet. When models excel on large-scale datasets but under-perform on specific tasks, selection strategy effectiveness drops, as shown in our experiments.

To this end, we introduce a novel paradigm to select and reuse VLMs called **M**odel **L**abel **L**earning (**MLL**). The core idea is to organize candidate pre-trained VLMs into a model hub and describe the specialty and utility of each VLM as the model's label in some manner. When faced with a new downstream task, we can match the task requirements with the model labels to select and reuse models. Specifically, the proposal contains three key interconnected modules: *model labeling*, *model selection*, and *model reuse*. In the *model labeling* process, we construct a semantic graph with commonly occurring visual concepts and representative samples, and each model undergoes pre-testing on the semantic graph to generate its model label, which describes its capability on these semantic classes. In the *model selection* process, we generate caption descriptions for both the nodes in the semantic graph and the categories to be classified in the target task to compare their similarity. This enables us to evaluate the model's performance on the target classes by aligning the matched semantic nodes with the model labels. In the *model reuse* process, we apply an ensemble strategy that combines the predictions of the selected models in a single class and chooses the highest confidence in all classes as the final prediction on the target task.

The model labeling process is completed immediately when the candidate VLM is added to the model hub, therefore, it is target task independent, which means the proposal is both

data and computationally efficient in the model selection process. Moreover, the proposal is highly growable since the capability could grow with the number of candidate models in the model hub and the model labels are also scalable since more semantic nodes can be added continually. To facilitate related research, we further introduce a comprehensive benchmark for evaluating VLM selection methods. The benchmark includes 49 pre-trained VLMs and 17 target datasets as downstream tasks. The ground-truth model ranking for each target task is provided for evaluation. We construct a semantic graph that contains more than 9000 commonly used visual concepts to pre-test VLMs. The experiments demonstrate the effectiveness of our approach in both selecting and reusing VLMs, while also validating the scalability of the model hub based on our proposal.

In summary, our contributions are as follows:

- **Problem:** We highlight that the performance of pre-trained VLM varies across different downstream tasks, even among classes within the same task. To address this, we first propose an important yet rarely studied problem called VLM selection and reuse, which will promote the deployment of VLM in more real tasks.

- **Method:** We propose a novel paradigm called Model Label Learning, which includes the processes of model labeling, selection, and reuse. This paradigm is both time- and data-efficient, and highly scalable. It can give birth to new VLM model hubs, which can simplifying user selection and reuse of VLMs for their tasks.

- **Evaluation:** We introduce a new benchmark for evaluating pre-trained VLM selection and reuse methods, advancing research in this field. Experimental results validate the effectiveness and scalability of our proposal for selecting and reusing VLMs.

## 2. Related Work

**Vision-Language Model.** In recent years, there have been significant advances in the field of Vision-Language Models (VLMs), including notable models such as CLIP (Radford et al., 2021), BLIP (Li et al., 2022), etc. These models leverage large-scale datasets containing image-text pairs, such as WIT (Srinivasan et al., 2021), to align visual and text features within a shared embedding space, which has led to impressive capabilities in feature extraction, particularly in the realm of zero-shot visual tasks. Tremendous works (Dosovitskiy et al., 2021; Yu et al., 2022; Fang et al., 2023) attempted to improve the zero-shot capabilities of VLMs by focusing on model architecture, pre-training datasets, and training/fine-tuning methods, which lead to the emergence of numerous open-source pre-trained VLMs. As a result, several VLM model hubs are constructed, such as open-clip (Ilharco et al., 2021) and HuggingFace (Wolf et al., 2020), which provide access to numerous VLMs. However, these model hubs lack effective model selection mechanisms; users cannot directly select suitable models for their tasks, they can only select models based on some quantitative indicators, such as download volume, popularity, etc.

**Model Selection.** As pre-trained models become increasingly diverse, how to select appropriate pre-trained models to tackle specific tasks has become a significant challenge. Many researchers have started to focus on this aspect. For example, Negative Conditional Entropy (NCE) (Tran et al., 2019) proposes an information-theoretic quantity to learn the transferability and hardness between classification tasks; LEEP (Nguyen et al., 2020) utilizes source prediction probabilities instead of hard labels compared with NCE; LogME (You et al., 2021) estimates the correlation between source model features and the target outputs by maximum evidence; MetaGL (Park et al., 2023) solves the model selection problem on graph data by introducing a meta-learning method; EMMS (Meng et al., 2023) uses weighted linear regression to estimate the transferability of candidate models; Model Spider (Zhang et al., 2024) uses a re-ranking mechanism to enhance the task-model co-embedding. Although these methods achieve well-performing in different settings, most of them focus on single-modal which cannot be directly used for VLM selection. Moreover, the training data for VLM is inaccessible, which introduces more challenges. Model selection for VLM is still a relatively new topic. LOVM (Zohar et al., 2023) uses a text dataset to describe the prediction task to train a linear model to predict the performance of the VLM. However, this method can only exploit text information and becomes less effective when there is a domain shift between downstream tasks and training tasks.

**Learnware.** Learnware (Zhou & Tan, 2022) is a novel paradigm that explores more effective model selection by

constructing specifications to describe the capabilities of the model, closely aligning with our idea of model labeling. Compared with previous selection methods, learnware enables scalable and efficient model selection across diverse architectures and input types within a unified framework, improving as the system expands. Model specification is central to the learnware paradigm, describing the functionality of each model. Recent works (Tan et al., 2024) on learnware paradigm are built on Reduced Kernel Mean Embedding (RKME) (Wu et al., 2021), which maps training data distributions to points in Reproducing Kernel Hilbert Space (RKHS) and identifies models by calculating the RKHS distance between RKME specifications. Furthermore, Guo et al. (2023) enhanced RKME for heterogeneous label spaces, while Tan et al. (2023) addressed challenges in heterogeneous feature spaces. Besides, Zhou et al. (2023) extended the learnware paradigm to the domain of the diffusion models selection. However, learnware requires training data to construct specifications. Considering the scale of VLM pre-trained datasets, it is unrealistic to construct specifications for learnware to select models due to limited time and computational resources.

## 3. Preliminaries

### 3.1. Zero-Shot Vision Task of VLM

Pre-trained VLMs for zero-shot visual tasks are built using two encoders: image encoder and text encoder. The image encoder is used to transform an image into a vector embedding, which presents its feature. The text encoder tokenizes the text input and generates a embedding representation by the text token. Let $\mathcal{I} : \mathcal{X} \to \mathbb{R}^n$ denotes the image encoder and $\mathcal{T} : \mathcal{Y} \to \mathbb{R}^n$ denotes the text encoder, where $X \in \mathcal{X}$ is the image input, $Y \in \mathcal{Y}$ is the text input, and $n$ is the dimension of the shared multi-modal embedding space of text embeddings and image embeddings.

In a particular downstream task $T$, there are $C_T$ classes $Y_T = \{y_i\}_{i=1}^{C_T}$. For a image $x \in X$, we obtain the image embeddings $\mathcal{I}(x)$ given by the image encoder $\mathcal{I}$ and the text embeddings $\mathcal{T}(y)$ of class $y$ produced by the text encoder $\mathcal{T}$. Then, the prediction $\hat{y}$ of image $x$ can be obtained as

$$\hat{y} = \arg\max_{y \in Y_T} \frac{\exp\left(\text{sim}\left(\mathcal{I}(x), \mathcal{T}(y)\right)/\tau\right)}{\sum\limits_{y' \in Y_T} \exp\left(\text{sim}\left(\mathcal{I}(x), \mathcal{T}(y')\right)/\tau\right)} \quad (1)$$

where $\text{sim}\left(\cdot, \cdot\right)$ denotes cosine similarity, $\tau$ represents the temperature determined by VLMs.

### 3.2. Problem Setup

Assume the model hub has $\mathcal{M}$ pre-trained VLMs $\{f_m = \{\mathcal{I}_m, \mathcal{T}_m\}\}_{m=1}^{\mathcal{M}}$, where $\mathcal{I}_m$ denotes the image encoder of the VLM $f_m$ and $\mathcal{T}_m$ denotes the text encoder of

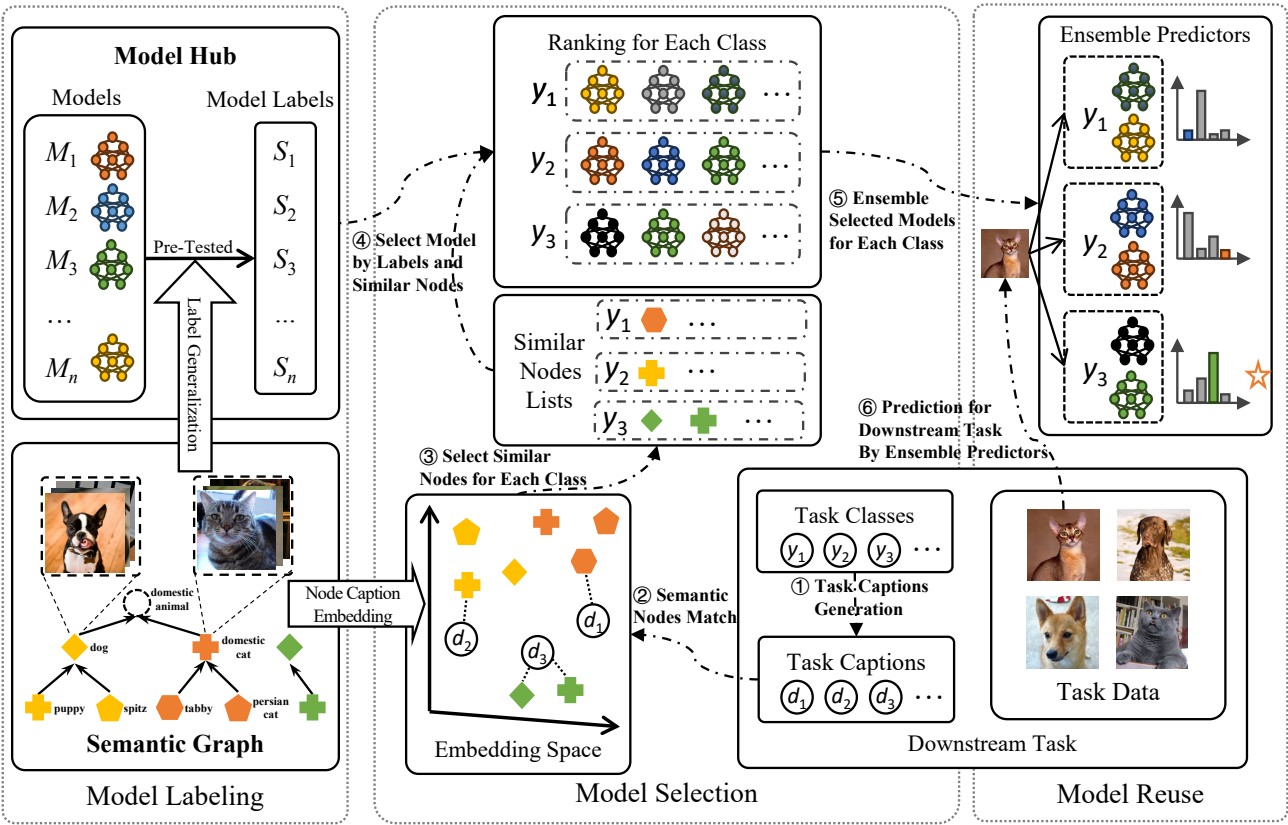

Figure 2. The framework of MLL paradigm. Models added to the hub first undergo a pre-testing phase, during which they are assigned labels that describe their specific functionalities in the labeling module. When a downstream task is presented, the system selects relevant models in the selection module and ensembles them to address the task.

the VLM $f_m$. There are two stages in our setting: *submission stage* for model developers to upload models to the model hub and *identification stage* for users to select suitable models for their tasks from the model hub.

In *submission stage*, model developers submit a VLM $f_m$ to the model hub, and the model hub assigns a label $S_m$ to the model to describe its specialty and utility. It is particularly emphasized that uploaded models are anonymous, meaning we do not have access to their training data.

In *identification stage*, user attempts to select VLMs from the model hub for the zero-shot downstream task $T$, by uploading general information about the task, such as classes, domain type, and task type, to describe their requirements. We subsequently utilize this information to select and reuse suitable VLMs, based on the model labels which have been established in the submission stage.

Two main problems in our settings are:

1) In *submission stage*, how can we design a label to fully characterize the capabilities of the submitted VLM?

2) In *identification stage*, how can we select and reuse

appropriate VLMs from the model hub to address users' downstream tasks based on their requirements and the model labels generated in the submission stage?

## 4. Our Approach

As illustrated in Figure 2, the MLL paradigm consists of three key modules: *model labeling*, *model selection*, and *model reuse*. In the *model labeling* process, MLL constructs a semantic graph $\mathcal{G}$ with commonly occurring visual concepts and representative samples as the evaluation datasets. When models are submitted to the model hub, they are pre-tested on the semantic graph and assigned labels $S_m$, which describe their capability on these semantic classes. In the *model selection* process, we generate caption descriptions for both the nodes in the semantic graph and the categories in the target task to compare their similarity. This enables us to evaluate the model's performance on the target classes by aligning the matched semantic nodes with the model labels. In the *model reuse* process, we apply an ensemble strategy that combines the selected models' predictions on a single class and chooses the highest confidence across all classes as the final prediction.

## 4.1. Model Labeling

To thoroughly characterize the capabilities of the model, we initiate the process by constructing a Semantic Graph $\mathcal{G}$ as evaluation datasets utilizing the WordNet (Miller, 1995) synsets. Firstly, we represent each synset in WordNet as a corresponding node $v$ within the semantic graph and establish links between nodes based on their relationships of hypernyms and hyponyms. Subsequently, to capture the real-world image distribution associated with each node, we randomly select images $X_v$ from sample datasets (detailed in Section 5.1) to serve as representations for each node $v$. Due to the limited information in synset name, we also need obtain the caption dataset $D_\mathcal{G} = \{d_v | v \in V_\mathcal{G}\}$ for label generalization where $V_\mathcal{G}$ denotes the set of nodes in Semantic Graph $\mathcal{G}$, $d_v$ denotes the caption of node $v$. We use "{synset name} which is {synset definition}" as the caption for each node, where "{synset name}" and "{synset definition}" correspond to the synset name and definition of a synset. Utilizing the constructed semantic graph, we generate a label $S_m$ for each VLM $f_m$ in the model hub that accurately reflects its capabilities.

$$s_{m,x}^v = \text{sim}(\mathcal{I}_m(x), \mathcal{T}_m(d_v)), x \in X_v \quad (2)$$

$$s_m^v = \{s_{m,x}^v | \ x \in X_v\} \quad (3)$$

$$S_m = \{s_m^v | \ v \in V_\mathcal{G}\} \quad (4)$$

where $\mathcal{I}_m(\cdot), \mathcal{T}_m(\cdot)$ denotes the image encoder and the text encoder of model $f_m$ .

Specifically, the constructed semantic graph allows for the seamless addition of new nodes and the incremental updating of model labels based on existing foundations. As the nodes in the semantic graph are expanded, its ability to reflect the performance capabilities of the models is enhanced. Once we have obtained labels for each model, we can utilize them for effective model selection.

## 4.2. Model Selection

In the model selection module, given a downstream task $T$ with $C_T$ classes $Y_T = \{y_i\}_{i=1}^{C_T}$, in order to utilize the obtained model labels $S_m$, we need to match the downstream task classes $Y_T$ with the semantic graph nodes $V_\mathcal{G}$. However, it can not match well using original class names. Inspired by previous work (Zohar et al., 2023), we construct expanded captions for both the downstream task classes and the semantic graph nodes. Large Language Models (OpenAI, 2023) have made significant advancements, facilitating the generation of text data. Assuming general information about the downstream tasks, such as task types and target domain, is accessible, we use GPT-4 with specific prompts to generate descriptions for each class as shown below, creating the caption dataset $D_T$ for downstream task $T$. The

following is an example of a prompt used to generate a caption of the class *cat*.

> Generate long detailed caption for the *natural picture* of *cat* in the *image classification*.
> e.g., "The *natural picture* of *cat*, which is ... ".
> Generate long caption for *cat* within *50* words.

where *natural picture* and *image classification* can be replaced with the domain and task descriptions, while *cat* can be substituted with the specific class name for target task.

Then, we can use a language model to generate embeddings of graph captions $D_\mathcal{G}$ and target task captions $D_T$. By comparing the cosine similarity between the embeddings, we can select the top $k$ nodes for each class based on similarity and construct a transfer matrix $Z = (z_{vy}) \in R^{|V^{\text{Selected}}| \times |Y^T|}$, where $V^{\text{Selected}}$ represents all selected nodes. Additionally, $z_{vy}$ represents the similarity of captions between graph node $v$ and task class $y$ if $v$ is among the top $k$ nodes that exhibit the highest similarity with task class $y$. Otherwise, it will be set to 0. Subsequently, the precision $p_{m,v}$ for each model $f_m$ at the graph nodes $v$ is defined as follows.

$$p_{m,v} = \frac{1}{|X_v|} \sum_{x \in X_v} \mathbb{I}\left(v = \arg\max_{v \in V^{\text{Selected}}} s_{m,x}^v\right) \quad (5)$$

By leveraging the transfer matrix $Z$, we can further derive the precision prediction $p_{m,y}$ for each class $y$ within the downstream task $T$ as described below.

$$p_{m,y} = \sum_{v \in V^{\text{Selected}}} p_{m,v} \cdot z_{vy} \quad (6)$$

When a model excels in a specific class, it may incorrectly handle data not belonging to that class. Consequently, we need to select models that perform well on specific classes while also maintaining good overall performance. Thus, we introduce a weight parameter $\alpha$ to balance class performance with overall performance. Then, the reuse metric $r$ for model $f_m$ in class $y$ is defined as:

$$r_{m,y} = \alpha \cdot p_{m,y} + \frac{1-\alpha}{|Y_T|} \sum_{y' \in Y_T} p_{m,y'} \quad (7)$$

## 4.3. Model Reuse

To better utilize the selection and harness the capabilities of models in the model hub, we introduce a specific count $k$ of models to reuse for each class $y$, we select up to $k$ highest-score model to form the ensemble predictor $\mathcal{F}_y^k = \{f_m \mid f_m \in \text{top-}k\left(\{r_{m,y}\}_\mathcal{M}\right)\}$. During testing, for the

**Algorithm 1** Model Selection & Reuse

---

**Input:** Model hub $\mathcal{M}$, model labels $\{S_m\}$, semantic graph $\mathcal{G}$, semantic graph caption dataset $D_{\mathcal{G}}$, count $k$ of reused models pre-class, target task $T = (X, Y)$
**Output:** Task prediction $\{\hat{y}\}$

1: Construct caption dataset $D_T$ for target task $T$.
2: Match similar nodes $V^{\text{Selected}}$ in $V_{\mathcal{G}}$ with $Y$ by captions $D_{\mathcal{G}}$ and $D_T$.
3: Construct transfer matrix $Z \in \mathbb{R}^{V^{\text{Selected}} \times C^T}$ based on caption similarity of $V^{\text{Selected}}$ and $Y$.
4: **for** $f_m \in \mathcal{F}_{\mathcal{M}}$ **do**
5:     Calculate reuse metric $r_{m,y}$ for each class $y$ in $Y$ by Eq.(5, 6, 7).
6: **end for**
7: **for** $y \in Y$ **do**
8:     Select $k$ models to ensemble predictor $\mathcal{F}_y^k = \left\{ f_m \mid f_m \in \text{top-}k \left( \{r_{m,y}\}_{\mathcal{M}} \right) \right\}$.
9:     Calculate prediction $\hat{y}$ for $x$ by Eq.(8, 9, 10).
10: **end for**
11: **Return** Task prediction $\{\hat{y}\}$.

---

data $x \in X$ of the downstream task, ensemble predictor $\mathcal{F}_y^k$ infers the confidence $p_y^k(x)$ of class $y$:

$$p_y^k(x) = \sum_{f_m \in \mathcal{F}_y^k} w_{m,y} \cdot \frac{\exp\left(\text{sim}\left(\mathcal{I}_m(x), \mathcal{T}_m(y)\right)\right)}{\sum_{y' \in Y_T} \exp\left(\text{sim}\left(\mathcal{I}_m(x), \mathcal{T}_m(y')\right)\right)} \tag{8}$$

where $w_{m,y}$ denotes the ensemble weight obtained from the output probability entropy $\mathcal{H}$ of each model within $\mathcal{F}_y^k$. $w_{m,y}$ is defined as:

$$w_{m,y} = \frac{\mathcal{H}\left(\{\text{sim}(\mathcal{I}_m(x), \mathcal{T}_m(y) \mid \forall\, y \in Y_T\}\right)}{\sum_{f_{m'} \in \mathcal{F}_y^k} \mathcal{H}\left(\{\text{sim}(\mathcal{I}_{m'}(x), \mathcal{T}_{m'}(y) \mid \forall\, y \in Y_T\}\right)} \tag{9}$$

Vishniakov et al. (2024) has shown VLMs predictions frequently exhibit overconfidence, which can degrade the performance of ensemble predictions, especially when the model makes incorrect predictions. To address this, we introduce $w_{m,y}$ based on the probability entropy $\mathcal{H}$ as a measure of uncertainty and assign lower weights to models with high confidence when they are overconfident, which reduces the impact of overconfidence, potentially erroneous predictions on the final ensemble output, thereby enhancing the robustness and accuracy of model's overall predictions.

Then, the class with the highest confidence is selected as the prediction $\hat{y}$ for $x$:

$$\hat{y}(x) = \arg\max_{y \in Y_T} p_y^k(x) \tag{10}$$

Flow of model selection and reuse of MLL Paradigm are summarized in Algorithm 1.

## 4.4. Discussion

Our proposal achieves higher accuracy, efficiency, and scalability. In terms of accuracy, the proposal elucidates the functionalities of VLMs by labeling models with a semantic graph that covers the most common visual concepts and representative samples to describe different data distributions, enabling more accurate identification of suitable models for users' target tasks. For efficiency, the proposal generates model labels when the pre-trained model is uploaded to the model hub, thus, it is highly efficient in the model selection process, without the need to run the candidate models on the target dataset. Regarding scalability, the concepts in the semantic graph can be continually added, thus, the model labels are scalable flexibility. Moreover, as the number of VLMs in the model hub increases, our proposal identifies higher-quality models, leading to improved performance on zero-shot downstream visual tasks.

# 5. Experiments

## 5.1. MLL Benchmark

To evaluate the capabilities of the MLL paradigm in zero-shot visual tasks with VLMs, we need to obtain a set of sampling datasets for constructing semantic graph $\mathcal{G}$, along with another set dedicated to downstream target tasks. For this study, we select 49 VLMs, 5 Sample Datasets, and 17 Target Datasets. Additionally, we collect general information about task types and domains associated with each dataset to provide a task description. For testing selected models on target tasks, we utilized same prompting strategy outlined in Radford et al. (2021), ensuring consistency in our evaluation methodology.

**Model Hub.** We leverage the open-clip library (Ilharco et al., 2021), which encompasses a diverse set of pre-trained VLMs across multiple architectural frameworks, such as ViT(Dosovitskiy et al., 2021) and ConvNet(Liu et al., 2022). These models have been pre-trained on a variety of large-scale datasets, such as WIT (Srinivasan et al., 2021) and LAION-2B (Schuhmann et al., 2022). We select 49 models from this library to form our model hub for the purpose of our experiments. All models used in the experiments are directed downloaded from the library.

**Datasets.** We utilized 5 datasets, ImageNet (Deng et al., 2009), ImageNet-V2 (Recht et al., 2019), ImageNet-Sketch (Wang et al., 2019), ImageNet-A (Hendrycks et al., 2021b) and ImageNet-R (Hendrycks et al., 2021a), as Sample Datasets for semantic graph construction. Additionally, we used 17 commonly used datasets and their task general information as Target Datasets to evaluate VLM selection and reuse methods in zero-shot visual tasks (as shown in Table 5). These datasets demonstrate diversity in terms of

*Table 1.* Comparison of the zero-shot performance on 17 downstream tasks by reusing a single model ($k = 1$). The best performance is highlighted in bold.

| Methods | CIFAR100 | Country211 | CLEVR-D | DTD | DMLab | Flowers102 | MNIST | OxfordPet | PCam |
|---|---|---|---|---|---|---|---|---|---|
| INB | 0.8599 | 0.3121 | 0.1262 | 0.6787 | 0.1940 | 0.8761 | 0.7956 | 0.9401 | **0.5332** |
| ModelGPT | 0.8599 | 0.3121 | 0.1262 | 0.6787 | 0.1940 | 0.8761 | 0.5648 | 0.9401 | 0.4990 |
| Proposal | **0.8773** | **0.3159** | **0.1361** | **0.6910** | **0.2111** | **0.8914** | **0.8101** | **0.9428** | 0.5003 |

| Methods | FER2013 | Food101 | GTSRB | RESISC45 | Rendered-SST2 | StanfordCars | STL10 | UCF101 | Avg. |
|---|---|---|---|---|---|---|---|---|---|
| INB | 0.2859 | 0.9553 | 0.5391 | 0.6139 | 0.5199 | 0.9487 | **0.9889** | 0.7702 | 0.6434 |
| ModelGPT | 0.4014 | 0.9553 | 0.5391 | 0.6139 | **0.5800** | 0.9487 | 0.9639 | 0.7702 | 0.6367 |
| Proposal | **0.4933** | **0.9566** | **0.5636** | **0.6437** | 0.5206 | **0.9568** | 0.9878 | **0.7961** | **0.6620** |

*Table 2.* Comparison of the zero-shot performance on 17 downstream tasks by ensemble 3 models per class ($k = 3$). The best performance is highlighted in bold.

| Methods | CIFAR100 | Country211 | CLEVR-D | DTD | DMLab | Flowers102 | MNIST | OxfordPet | PCam |
|---|---|---|---|---|---|---|---|---|---|
| INB | **0.8977** | 0.3228 | 0.1048 | 0.6968 | 0.1296 | 0.8873 | 0.7847 | 0.9433 | 0.5002 |
| ModelGPT | 0.8949 | **0.3243** | 0.0907 | 0.6957 | 0.1338 | **0.8876** | 0.7888 | **0.9490** | 0.5002 |
| Proposal | 0.8923 | 0.3238 | **0.1171** | **0.7053** | **0.1573** | 0.8720 | **0.8101** | 0.9428 | **0.5003** |

| Methods | FER2013 | Food101 | GTSRB | RESISC45 | Rendered-SST2 | StanfordCars | STL10 | UCF101 | Avg. |
|---|---|---|---|---|---|---|---|---|---|
| INB | 0.2636 | 0.9606 | 0.5938 | 0.6615 | **0.5332** | 0.9540 | 0.9949 | 0.7966 | 0.6498 |
| ModelGPT | 0.2653 | **0.9611** | **0.5980** | 0.6555 | 0.5299 | 0.9533 | **0.9957** | 0.7933 | 0.6480 |
| Proposal | **0.4933** | 0.9566 | 0.5636 | **0.6800** | 0.5233 | **0.9541** | 0.9854 | **0.8092** | **0.6639** |

domain, number of classes, and task types. They encompass various domains, including animals, food, text, landscapes, remote sensing, medical, and transportation. Additionally, they cover a range of tasks such as image classification, geo-localization, optical character recognition, facial expression recognition, land cover classification, and object distance estimation. To eliminate interference of predication from additional modules during evaluation, all tasks can be assessed using the same VLM architecture.

**Evaluation Metrics.** In our benchmark, methods are expected to select models from a hub of 49 pre-trained VLMs and reuse them across 17 target datasets as downstream tasks to achieve better performance. Notably, all models selected for use are without additional fine-tuning, as all downstream tasks are zero-shot. We use *Acc.* to evaluate methods' performance on both downstream target tasks and the average performance across all tasks.

### 5.2. Experiment Setup

**Semantic Graph Construction.** In order to build a rich semantic graph $\mathcal{G}$, we used WordNet synsets to carefully select 9055 nodes, covering many fields such as animals, tools, clothing, vehicles, plants, ensuring the representative-ness of the semantic graph. We randomly selected up to 75 pictures for each node, which are all from the sample datasets and reflected the distribution of the node's concepts. In addition, we used the OpenAI text-embedding-3-large model to obtain their caption embeddings. By calculating the cosine similarity between these embeddings, we can accurately match the correspondence between the semantic graph nodes and the downstream task classes.

**Compared Methods.** Initially, we compare our proposal with ImageNet Baseline (INB), which simply select the model which has best performance on the ImageNet in the model hub to reuse. Additionally, we compare it with a VLM selection method called ModelGPT (Zohar et al., 2023). ModelGPT employs generated captions and synonyms for target task classes as substitutes for images of those classes, then evaluates the performance of VLMs by measuring their ability to correctly classify the captions and synonyms into their corresponding classes, which serves as the reuse metric in combination with INB. A linear model is then learned between the reuse metric and ground-truth performance on training downstream tasks. Finally, the zero-shot ability of VLMs on the target task is predicted using this linear model and the reuse metric.

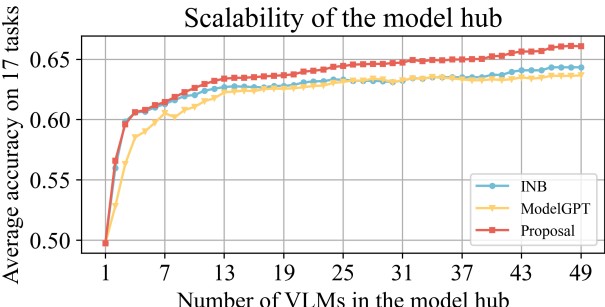

*Figure 3.* Average performance on 17 downstream tasks with the scaling of the model hub.

*Table 3.* Average performance of our proposal on 17 downstream tasks compared with different $\alpha$ by reusing single model ($k = 1$)

| $\alpha$ | 0.5 | 0.6 | 0.7 | 0.8 | 0.9 |
|---|---|---|---|---|---|
| Avg. | 0.6569 | 0.6571 | **0.6620** | 0.6600 | 0.6590 |

*Table 4.* Average Performance and Interface Time Cost of our proposal on 17 downstream tasks by different reusing model count $k$.

| $k$ | Average Accuracy | Interface Time Cost Compared with $k = 1$ |
|---|---|---|
| 1 | 0.6620 | 1.000 |
| 2 | 0.6637 | 1.927 |
| 3 | **0.6639** | 2.233 |
| 4 | 0.6571 | 2.467 |
| 5 | 0.6556 | 2.700 |
| 6 | 0.6594 | 2.900 |

**Implementation Details.** We adopt the official code to implement ModelGPT. For a fair comparison, the experiment utilizes the ground-truth performance of VLMs on Sample Datasets for ModelGPT to train its linear model, and then evaluate it on the benchmark. For both INB and ModelGPT, the experiment selects the model with the highest predictive performance given by the method for reuse in the target task. Specifically, we employ the same prompting strategy outlined in Radford et al. (2021), which uses the prompt "a photo of {class}", where "{class}" is replaced by the task class. All selected models are utilized without any further fine-tuning, given that all downstream tasks are conducted in a zero-shot manner. Additionally, the weight $\alpha$ for model selection in our setting is set to 0.7. All experiments are conducted on NVIDIA A800 GPUs.

### 5.3. Experiment Results

**Zero-shot Performance.** In our setup, the goal is to optimize the performance of VLMs on downstream zero-shot visual tasks. Therefore, in Table 1 and Table 2, we compare the performance of different model selection methods across 17 target datasets. We set two values for the count $k$ of reused models, specifically 1 and 3, to test the effects of using a single model versus an ensemble of three models per class. In the ensemble setup, we first selected the top-$k$ models from each method and then applied the same reuse strategies discussed in Section 4.4. The results show that our method achieves high performance on most downstream tasks. **ModelGPT largely aligns with INB, indicating a strong correlation in their selection strategies. When INB fails to select a well-performing model, ModelGPT also struggles with selection**. By comparing different counts $k$ of reused models, MLL demonstrates that reusing the model with the best performance per class is often sufficient to outperform baseline methods in most downstream tasks, highlighting the practicality of the MLL paradigm. We also find that in datasets with a limited number of classes, such as PCam, employing a single model for

each class tends to yield better results. Additionally, when the models available in the model hub are generally weak, as seen in several datasets, such as CLEVR-D, relying on ensemble methods may introduce more noise than benefit. In these cases, a single model per class often provides the ultimate balance between simplicity and effectiveness.

**Scalability of Model Hub.** We design a scenario where the model hub starts from scratch and gradually expands until it contains all available VLMs. Figure 3 provides a detailed illustration of the average performance of 17 downstream tasks throughout 30 randomly generated expansion schemes. The results show that as the model hub grows, our method can more efficiently reuse the well-performing VLM models for various tasks, reducing the limitations in model selection and boosting system performance across a range of visual tasks. This shows that our method is not only highly effective in the present but also holds the potential for continued improvement as the model hub grows.

**Ablation Study.** To analyze the influence of key hyperparameters in our proposal, we conducted an ablation study focusing on the effects of the parameter $\alpha$ (as defined in Equation 7) and the reuse model count $k$. The impact of $\alpha$ is presented in Table 3. The results demonstrate that our proposal consistently outperforms the compared methods across a range of $\alpha$ values, highlighting the robustness of the model to this parameter. Additionally, the effect of $k$ is shown in Table 4, where the results show that using a smaller number of models strikes an optimal balance between performance and computational efficiency, with minimal performance loss compared to larger ensembles.

## 6. Conclusion

In this paper, we explore how to select and reuse pre-trained VLMs for a specific downstream task. To the best of our knowledge, this problem has rarely been studied. To address this, we propose a novel paradigm called **M**odel **L**abel **L**earning (MLL) that assigns each VLM a label to describe its utility on representative visual concepts. The MLL paradigm contains three key modules: *model labeling*, *model selection*, and *model reuse*. The proposal is highly efficient, scalable, and convenient for both model developers and users. Moreover, we introduced a benchmark for evaluating pre-trained VLM selection and reuse methods that contain 49 pre-trained VLMs and 17 target datasets. Experiments demonstrate that the proposal can achieve state-of-the-art VLM selection performance, and the ability to deal with downstream tasks could grow with the scale of the model hub, showing the potential of building large model hubs with advanced model selection mechanisms.

In future work, we will endeavor to develop a novel model hub based on the MLL paradigm presented in this paper, allowing valid VLM developers from all over the world to submit their models. When users work on visual tasks, they will be able to select and reuse models from the hub. The limitation of this paper is that the current implementation focuses solely on VLMs and visual classification tasks. We will further attempt to extend our paradigm to more model types that have significant architectural differences compared with VLMs, and more complex tasks.

## Code Availability Statement

The implementation code of benchmark and our proposal for the MLL paradigm is available at https://github.com/LAMDASZ-ML/MLL.

## Acknowledgements

This research was supported by National Science Foundation of China (62306133, 62250069) and Key Program of Jiangsu Science Foundation (BK20243012). We would like to thank the reviewers for their constructive suggestions.

## Impact Statement

This paper aims to explore how to select and reuse pre-trained VLMs for a specific downstream task. We are convinced that the majority of the impacts are positive, which is not necessary to specifically highlight any particular impacts in this paper. Moreover, we expect that the responsible use of technology will naturally promote discussions around best practices and regulations for implementing methods.

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

# A. Details of Benchmark

In this section, we provide detailed insights into our benchmark utilized for evaluating VLM selection and reuse methods. Table 5 outlines the datasets used in the benchmark, highlighting the type of domain and task for each dataset. This breakdown is essential for understanding the context and effectiveness of the models assessed in our study. Table 6 presents general information on the model hub, including model architecture, pre-trained datasets, parameters, FLOPs, and accuracy on ImageNet.

*Table 5.* Details on the datasets used in the benchmark, which contain the type of domain and task.

| Dataset | Domain | Task |
| --- | --- | --- |
| ImageNet (Deng et al., 2009) | natural picture | image classification |
| ImageNet-V2 (Recht et al., 2019) | natural picture | image classification |
| ImageNet-Sketch (Wang et al., 2019) | Sketch picture | image classification |
| ImageNet-A (Hendrycks et al., 2021b) | natural picture | image classification |
| ImageNet-R (Hendrycks et al., 2021a) | 15 domain picture (e.g., art, cartoon) | image classification |
| CIFAR100 (Alex, 2009) | natural picture | image classification |
| Country211 (Radford et al., 2021) | natural picture | geo-localization |
| CLEVR-D (Johnson et al., 2017) | natural picture | object distance estimation |
| DTD (Cimpoi et al., 2014) | texture picture | image classification |
| DMLab (Zhai et al., 2019) | natural picture | object distance estimation |
| Flowers102 (Nilsback & Zisserman, 2008) | flower picture | image classification |
| FER2013 (Goodfellow et al., 2013) | facial picture | facial expression classification |
| Food101 (Bossard et al., 2014) | food picture | image classification |
| GTSRB (Stallkamp et al., 2012) | traffic picture | image classification |
| MNIST (LeCun et al., 1998) | digit picture | image classification |
| OxfordIIITPet (Parkhi et al., 2012) | pet photograph | image classification |
| PCam (Veeling et al., 2018) | medical picture | image classification |
| Rendered SST2 (Radford et al., 2021) | text picture | optical character recognition |
| RESISC45 (Cheng et al., 2017) | satellite picture | land cover classification |
| StanfordCars (Krause et al., 2013) | car picture | image classification |
| STL10 (Coates et al., 2011) | natural picture | image classification |
| UCF101 (Soomro et al., 2012) | video frame | action recognition |

*Table 6.* Details on model hub used in the benchmark, which contain the model architecture, pre-trained datasets, parameters, FLOPs, and Accuracy on ImageNet

| ID | Model Architecture | Pretrained Dataset | Params (M) | FLOPs (B) | ImageNet Acc. |
|---|---|---|---|---|---|
| 1 | RN50 | openai | 102.01 | 18.18 | 0.5982 |
| 2 | RN50 | cc12m | 102.01 | 18.18 | 0.3591 |
| 3 | RN101 | openai | 119.69 | 25.5 | 0.6228 |
| 4 | RN101 | yfcc15m | 119.69 | 25.5 | 0.3407 |
| 5 | RN101-quickgelu | openai | 119.69 | 25.5 | 0.6228 |
| 6 | RN101-quickgelu | yfcc15m | 119.69 | 25.5 | 0.3487 |
| 7 | RN50x4 | openai | 178.3 | 51.82 | 0.6627 |
| 8 | RN50x64 | openai | 623.26 | 552.65 | 0.7391 |
| 9 | ViT-B-32 | openai | 151.28 | 14.78 | 0.6332 |
| 10 | ViT-B-32 | laion2b_e16 | 151.28 | 14.78 | 0.6565 |
| 11 | ViT-B-32 | datacomp_xl_s13b_b90k | 151.28 | 14.78 | 0.6917 |
| 12 | ViT-B-32 | commonpool_m_clip_s128m_b4k | 151.28 | 14.78 | 0.2725 |
| 13 | ViT-B-32-256 | datacomp_s34b_b86k | 151.29 | 17.46 | 0.7281 |
| 14 | ViT-B-32-quickgelu | laion400m_e31 | 151.28 | 14.78 | 0.6294 |
| 15 | ViT-B-32-quickgelu | metaclip_fullcc | 151.28 | 14.78 | 0.6766 |
| 16 | ViT-B-16 | openai | 149.62 | 41.09 | 0.6834 |
| 17 | ViT-B-16 | laion2b_s34b_b88k | 149.62 | 41.09 | 0.7023 |
| 18 | ViT-B-16 | datacomp_l_s1b_b8k | 149.62 | 41.09 | 0.6310 |
| 19 | ViT-B-16 | commonpool_l_laion_s1b_b8k | 149.62 | 41.09 | 0.5526 |
| 20 | ViT-B-16 | dfn2b | 149.62 | 41.09 | 0.7624 |
| 21 | ViT-B-16-quickgelu | metaclip_fullcc | 149.62 | 41.09 | 0.7212 |
| 22 | ViT-B-16-plus-240 | laion400m_e31 | 208.38 | 64.03 | 0.6904 |
| 23 | ViT-L-14 | openai | 427.62 | 175.33 | 0.7554 |
| 24 | ViT-L-14 | laion400m_e31 | 427.62 | 175.33 | 0.7271 |
| 25 | ViT-L-14 | datacomp_xl_s13b_b90k | 427.62 | 175.33 | 0.7921 |
| 26 | ViT-L-14 | commonpool_xl_clip_s13b_b90k | 427.62 | 175.33 | 0.7637 |
| 27 | ViT-L-14-quickgelu | metaclip_fullcc | 427.62 | 175.33 | 0.7917 |
| 28 | ViT-L-14-quickgelu | dfn2b | 427.62 | 175.33 | 0.8141 |
| 29 | ViT-L-14-336 | openai | 427.94 | 395.22 | 0.7656 |
| 30 | ViT-H-14 | laion2b_s32b_b79k | 986.11 | 381.68 | 0.7796 |
| 31 | ViT-H-14-quickgelu | metaclip_fullcc | 986.11 | 381.68 | 0.8051 |
| 32 | ViT-H-14-378-quickgelu | dfn5b | 986.71 | 1054.05 | 0.8437 |
| 33 | ViT-g-14 | laion2b_s12b_b42k | 1366.68 | 581.15 | 0.7663 |
| 34 | ViT-bigG-14 | laion2b_s39b_b160k | 2539.57 | 1065.36 | 0.8009 |
| 35 | roberta-ViT-B-32 | laion2b_s12b_b32k | 212.72 | 105.87 | 0.6171 |
| 36 | xlm-roberta-base-ViT-B-32 | laion5b_s13b_b90k | 366.12 | 105.87 | 0.6236 |
| 37 | convnext_base_w | laion2b_s13b_b82k | 179.39 | 49.38 | 0.7078 |
| 38 | convnext_base_w_320 | laion_aesthetic_s13b_b82k | 179.39 | 71.94 | 0.7167 |
| 39 | convnext_large_d | laion2b_s26b_b102k_augreg | 351.77 | 107.5 | 0.7591 |
| 40 | convnext_large_d_320 | laion2b_s29b_b131k_ft | 351.77 | 157.98 | 0.7660 |
| 41 | convnext_xxlarge | laion2b_s34b_b82k_augreg_soup | 1200.58 | 443.03 | 0.7947 |
| 42 | coca_ViT-B-32 | laion2b_s13b_b90k | 253.56 | 33.34 | 0.6331 |
| 43 | coca_ViT-L-14 | laion2b_s13b_b90k | 638.45 | 214.52 | 0.7561 |
| 44 | EVA01-g-14 | laion400m_s11b_b41k | 1136.44 | 547.36 | 0.7852 |
| 45 | EVA02-B-16 | merged2b_s8b_b131k | 149.69 | 41.09 | 0.7472 |
| 46 | EVA02-L-14-336 | merged2b_s6b_b61k | 428.08 | 395.16 | 0.8039 |
| 47 | EVA02-E-14 | laion2b_s4b_b115k | 4704.59 | 2311.42 | 0.8196 |
| 48 | nllb-clip-base | v1 | 501.89 | 369.6 | 0.2432 |
| 49 | nllb-clip-base-siglip | v1 | 507.47 | 472.91 | 0.3909 |

