# OpenReview forum: "Vision-Language Model Selection and Reuse for Downstream Adaptation"
_ICML.cc/2025/Conference — ICML 2025 poster_

### Official Review · Reviewer_9kf2 · 2025-02-19

**Overall Recommendation:** 2

**Summary:**

The paper deals with Model Label Learning (MLL) to select and reuse pre-trained VLMs for specific downstream tasks. It addresses the challenge of choosing the best VLMs from numerous options, each with varying performance on different tasks and different classes. The MLL approach includes model labeling, selection, and reuse, utilizing a predefined concept pool to bridge the understanding of those concepts from candidate VLMs and target tasks.

**Claims And Evidence:**

1. The contribution of this work appears to be weak. The so-called "new benchmark" is not a separate contribution, as it is inherently tied to the task or paradigm presented. Therefore, they should be considered a single contribution rather than two distinct ones.
2. The validation and analysis provided in the paper are weak (see Experimental Designs Or Analyses). Additionally, the improvements demonstrated by the proposed model are limited: In the best set of experiments (0.6639, k=3), the author proposed method outperformed the comparison method in only half of the downstream tasks. This raises concerns about the effectiveness of the proposed model selection method.

**Essential References Not Discussed:**

Given that the authors present the proposed paradigm as one of their main contribution, it is essential to thoroughly evaluate its effectiveness compared to other training-free methods for VLMs, such as but not limited to test-time adaptation.

**Experimental Designs Or Analyses:**

The submission has not been well validated, and analysis is weak and not insightful enough. Besides above mentioned weakness,
1. important baselines are missing. For example, since ensemble typically enhances performance and random sampling can ensure diversity in recognizing different classes, it is crucial to conduct random ensemble with majority voting / prob. distribution average / etc as ablation study, other than the parameter analysis presented in Section 5.3.
2. how do the randomly selection reference images affect the robustness of the proposed method?
3. how do specific downstream tasks correlate with different model candidates? For example, why does the method show significant improvement on datasets DMLab, RESISC45 and FER2013, while performing at a comparable level on other datasets?

**Methods And Evaluation Criteria:**

The process of constructing the reference dataset, profiling the target dataset, and conducting multi-model inference, replying on a combination of language generation models, embedding models, and similarity computation, is complex and time-consuming. As can be seen in Tab.2, a suboptimal choice of k can lead to a decline in results.
Plus the details of this experiment are missing, it is important to question how robust the selection of k=3 is across 17 different downstream tasks. If the value of k is uncertain unless given a validation split, it becomes challenging to demonstrate how effective the procedures could be in practical applications.

**Other Comments Or Suggestions:**

Please see comments above.

**Other Strengths And Weaknesses:**

While the importance of the task is acknowledged, the presented method has not been convincingly validated and appears to be impractical.

**Questions For Authors:**

How is X_v determined for each v that is randomly collected?  Is there a one-to-one correspondence for each v, as defined by WordNet, across all 5 datasets?

**Relation To Broader Scientific Literature:**

The proposed method, while simple and straightforward, lacks convincing validation and practical applicability. The method's effectiveness is limited even after model ensemble. It also raises important questions about the robustness and practicality of its approach, especially considering the complexity and uncertainty involved in key processes like target benchmark profiling, model selection and dataset correspondence. These factors suggest that while the paper addresses a relevant and meaningful task, its contributions may not significantly advance the field without further validation and refinement.

**Theoretical Claims:**

The proposed method is straightforward and, for the most part, clearly explained.
Two minor suggestions:
1. The logic from eq6 to eq7 is hard to follow.
2. \mathcal{T}_m(D) should be \mathcal{T}_m(d) in eq2.

---

### Official Review · Reviewer_Vdth · 2025-03-11

**Overall Recommendation:** 4

**Summary:**

This paper presents a framework for organizing models that facilitates the storage, labeling, and reuse of vision-language models (VLMs). This system enhances overall performance compared to using a single VLM. A model labeling process is introduced to precisely describe the functionality of each VLM, enabling subsequent identification and reuse. The authors propose a comprehensive benchmark involving 49 VLMs and 17 target tasks to evaluate and demonstrate the effectiveness of the proposed method.

**Claims And Evidence:**

The authors claim that improved performance can be achieved with the proposed model organization framework because different Vision-Language Models (VLMs) have their own advantages in handling various models, and the proposed method can fully utilize these advantages. The experiments conducted in this paper have thoroughly demonstrated the authors' claim. They not only show that the proposed method achieves better overall performance on 17 tasks but also indicate that as the number of models increases, the overall performance can further improve.

**Essential References Not Discussed:**

No necessary reference is omitted.

**Experimental Designs Or Analyses:**

Both the experimental design and analyses are comprehensive. The experiments are conducted on multiple datasets with up-to-date comparison methods. Moreover, the authors give scalability and ablation studies to further demonstrate the effectiveness of proposed methods.

**Methods And Evaluation Criteria:**

The proposed method seems to be work. To evaluate the performance of the method, the authors introduce a semantic graph with labels and corresponding images to pre-test the performance of each model. Subsequently, when reusing each model, the authors propose a model identification strategy to locate each model in an embedding space and ensemble models for subsequent tasks. The overall framework has potential to demonstrate the effectiveness of a collection of models. The benchmark and evaluation in this paper are valid and can reflect the performance of each method.

**Other Comments Or Suggestions:**

Please see the strengths and weaknesses above.

**Other Strengths And Weaknesses:**

Strengths:

1.	The problem addressed in this paper is significant and intriguing. Managing large Vision-Language Models (VLMs) in the model hub is crucial because training new models for new downstream tasks is costly, whereas existing models can be utilized to handle these tasks.
2.	The proposed method is logical, as the pipeline of pre-testing, model identification, and model reuse appears effective. With a more comprehensive semantic graph and an increased number of models in the model hub, this method can achieve better performance.
3.	The experiments in this paper are thorough. The authors have developed a benchmark with 49 VLMs and 17 tasks, which thoroughly evaluates the performance of each method. The proposed method achieves the best overall performance, demonstrating the effectiveness of the proposed framework.

Weakness:

The primary limitation of this paper is that the model hub is not sufficiently large to fully enhance the effectiveness of the proposed method. Nevertheless, I believe the current benchmark is adequate to assess the effectiveness of each method and demonstrate the proposal's effectiveness. In this context, I do not perceive any significant weaknesses in the paper.

**Questions For Authors:**

- Have you made the computational cost analysis for the proposed method? Is the proposed method more efficient or slower than the previous works?

**Relation To Broader Scientific Literature:**

The method presented in this paper is significant, as it can be utilized to manage existing Vision-Language Models (VLMs) to address new downstream tasks with improved performance, rather than constructing new VLMs, which is costly. These techniques can assist platforms like HuggingFace in better organizing models.

**Theoretical Claims:**

N/A

---

### Official Review · Reviewer_Hxkz · 2025-03-13

**Overall Recommendation:** 3

**Summary:**

The paper explores a practical VLM reuse problem and proposes Model Label Learning (MLL), an efficient approach for selecting and reusing pre-trained Vision-Language Models (VLMs) for downstream tasks. The framework comprises three modules: (1) Model Labeling, which assigns labels to VLMs based on their capabilities; (2) Model Selection, which matches these labels to task requirements; and (3) Model Reuse, which employs an ensemble of selected models. Additionally, a large-scale benchmark, including 49 VLMs and 17 datasets, is introduced to evaluate MLL’s effectiveness. Experimental results demonstrate its promising scalability and performance.

**Claims And Evidence:**

The paper claims that the performance of VLMs can vary significantly depending on the dataset domain. To support this claim, the authors conduct extensive experiments, as shown in Figure 1. Specifically, Figure 1(a) illustrates that VLMs exhibit different strengths across various visual tasks, with no single model consistently outperforming all others across every task. Figure 1(b) demonstrates that even within the same task, different VLMs achieve varying levels of performance across specific classes.

**Essential References Not Discussed:**

NA

**Experimental Designs Or Analyses:**

The paper presents comprehensive experiments involving 49 VLMs and 17 target task datasets, along with sufficient analysis. The experimental results demonstrate the effectiveness of the proposed method in selecting and reusing VLMs.

In addition to the experiments mentioned above, the authors should provide more experimental results to further validate the proposed method. For example, for each target dataset, the highest performance achieved by any model in the model hub should be included as a reference. This would help assess the effectiveness of the proposed method in selecting models.

**Methods And Evaluation Criteria:**

To achieve effective reuse of VLMs, the paper proposes Model Label Learning (MLL), an efficient approach for selecting and reusing pre-trained Vision-Language Models (VLMs) for downstream tasks. Specifically, the framework consists of three modules: (1) Model Labeling, which assigns labels to VLMs based on their capabilities; (2) Model Selection, which matches these labels to task requirements; and (3) Model Reuse, which employs an ensemble of selected models. The proposed framework is reasonable for addressing this problem.

Additionally, a large-scale benchmark, including 49 VLMs and 17 datasets, is introduced to evaluate the effectiveness of MLL.

**Other Comments Or Suggestions:**

In addition to VLMs for image classification tasks, more VLM studies have explored dense-level recognition tasks, such as detection and segmentation. The authors could consider incorporating more types of VLMs into the proposed framework, enabling it to handle a wider range of tasks.

**Other Strengths And Weaknesses:**

The main strength of this work is that the problem it explores is both practical and meaningful. Additionally, it demonstrates good scalability, as the use of a semantic graph enables MLL to expand as new models or tasks are added, making it adaptable to diverse visual tasks. The main weakness is that more experimental results should be provided. Please refer to Experimental Designs Or Analyses for details.

**Questions For Authors:**

Please refer to above sections for details.

**Relation To Broader Scientific Literature:**

Unlike other VLM studies, this paper explores a practical VLM reuse problem, aiming to effectively reuse existing models to improve performance on target datasets.

**Theoretical Claims:**

NA

---

### Official Review · Reviewer_zHkU · 2025-03-16

**Overall Recommendation:** 4

**Summary:**

This paper introduces Model Label Learning (MLL) for selecting and reusing pre-trained VLMs for downstream tasks. This method aims to address the challenge of choosing the best VLM from a growing hub, given their diverse performance across tasks and primarily it is impractical to evaluate them exhaustively. The core idea of MLL is to assign "model labels" to each VLM, describing its capabilities based on pre-testing on a semantic graph of visual concepts. MLL then utilizes these labels to efficiently select and reuse VLMs for new tasks by semantically matching task requirements to model labels. The MLL paradigm consists of three modules: model labeling (pre-testing VLMs on a semantic graph and creating labels), model selection (matching task descriptions to model labels for VLM selection), and model reuse (ensembling selected VLMs). In addition, the authors introduce a new benchmark for VLM selection, comprising 49 VLMs and 17 downstream datasets. Experimental results on this benchmark show the effectiveness of MLL compared to baselines like ImageNet Baseline (INB) and ModelGPT, demonstrating improvement in zero-shot downstream tasks.

## update after rebuttal

My questions have been addressed by the rebuttal and I have thus increased my score.

**Claims And Evidence:**

### C1: MLL is effective for selecting and reusing VLMs for downstream tasks, leading to improved performance.
- E1: Tables 1 and 2 show improved average accuracy and F1-score compared to INB and ModelGPT across 17 downstream tasks for both single model (k=1) and ensemble (k=3) settings. This provides initial evidence. Then, Table 3, 4 demonstrate the robustness to hyper-parameters.
- P1: I think this claim might be strengthened by including statistical significance test. It would help determine whether the average improvement across 17 datasets (1.86% over INB, 2.53% over ModelGPT) represents a meaningful pattern.

### C2: MLL is computationally efficient in the model selection process.
- E2: The paper states that model labeling is target task independent and pre-computed, making the selection process efficient. This is a design feature suggesting methodology efficiency.
- P2: However, empirical evidence of the actual computational time for model selection and comparison to baselines is not provided  (although Table 4 hints at interface time costs). The cost of labeling stage itself is not quantified.


### C3: MLL is scalable as the model hub grows.
- E3: Figure 3 shows that average performance generally increases as the number of VLMs in the hub grows, suggesting scalability.
- P3: The rate of performance improvement with hub size and the saturation point are not fully explored. The computational cost of maintaining and updating model labels as the hub grows is not discussed. Moreover, the experiment in Figure 3 is based on random expansion, which might not fully reflect real-world hub growth scenarios, such as chronological additions based on when models are published, preferential inclusion of higher-performing models first, or somehow strategic additions based on identified performance gaps. I personally think that the random expansion approach does not capture these realistic growth patterns, this likely matters because the order in which models are added could significantly affect the performance improvement curve. For instance, if high-performing models happen to be added early in many random schemes, this could show rapid initial improvements that might not be achievable in practice.

In brief, the claims made are initially supported by the presented data, but it is better to be strengthened with more empirical evidence, statistical validation, and more detailed analyses of costs and scalability.

**Essential References Not Discussed:**

Primarily I don't think there are significant missing of essential literature; however, to strengthen the literature context and ensure comprehensiveness, the authors might consider discussing and citing works in the following more specific areas:

- semantic similarity measures: as MLL relies on semantic similarity measure for comparing captions, discussing relevant papers on semantic textual similarity and different metrics would be beneficial.
- ensemble methods for VLMs: If the ensemble reuse module could be extended to more ensembling techniques, making a discussion on relevant works on ensemble methods, would also be important.

**Experimental Designs Or Analyses:**

The experimental designs and analyses are sound in principle but require more detailed reporting and validation.

#### Soundness
Using INB and ModelGPT as baselines is appropriate and allows for comparison to a simple baseline and a relevant existing VLM selection method. The use of a diverse benchmark is a strength. The ablation studies on $\alpha$ (table 3) and model count $k$ (table 4) demonstrate an effort to understand the impact of key hyperparameters and design choices.

#### Issues and areas for improvements:
1. statistical significance: as aforementioned, this submission lacks mention of significance test to validate the observed performance differences.
2. more detailed result: while average performance is reported, the readers might also wonder: are the improvements consistent across all datasets? Also, are there specific datasets or task types where MLL excels or underperforms?
3. computational cost: the analysis of the cost of model labeling should also be reported.

**Methods And Evaluation Criteria:**

The proposed methods and evaluation criteria are generally sensible and appropriate for the problem of VLM selection and reuse.

The MLL method with its three modules is a logical approach to address the VLM selection problem. Semantic labeling provides a structuredway to understand VLM capabilities without task-specific evaluation. Semantic matching for selection is an efficient alternative to exhaustive search. Ensemble reuse leverages the strengths of multiple selected models.

The introduced benchmark of 49 VLMs and 17 downstream datasets is a strong evaluation criterion, covering a range of domains and tasks relevant to VLMs. The authors propose to use accuracy and F1-score, which are standard metrics for the evaluated tasks. Comparison against INB and ModelGPT provides relevant baselines.

**Other Comments Or Suggestions:**

I have no more comments except for the existing sections.

**Other Strengths And Weaknesses:**

I summarize the strengths and weaknesses I identified below. They may overlap with the previously discussed points:

### strengths

1. The paper addresses a practical problem that will become increasingly important as more pre-trained VLMs become available.
2. The MLL method is conceptually simple, with clearly separated components for labeling, selection, and reuse.
3. The approach is computationally efficient for users, as the potentially expensive model labeling process occurs only once when models are added to the hub
4. The benchmark will be valuable for future research in VLM selection

### Weaknesses:
1. The approach depends heavily on the the quality of the constructed semantic graph, but the graph construction process isn't explored in depth.
2. The computational overhead of the model labeling process itself isn't thoroughly discussed, which could be significant for large models
3. More empirical results (e.g., statistical significance tests) should be reported as well.
4. Analysis of failure cases where MLL does not perform very well would also be meaningful.

**Questions For Authors:**

1. how sensitive is the approach to the choice of nodes in the semantic graph, and would a different semantic graph graph construction lead to significantly different results?
2. What is the computational overhead of the model labeling process, particularly for large models?
3. How would the framework handle non-classification tasks?
4. The entropy-based ensemble weighting scheme is relevant to model overconfidence, but have you explored other weighting strategies?

**Relation To Broader Scientific Literature:**

This paper appears to be relating to three research areas:

VLM: The paper directly relates to a challenge in VLMs - how to efficiently and effectively utilize the increasing number of pre-trained models.

Model Selection: The paper also relates to literature on model selection, particularly for pre-trained models. It acknowledges and contrasts with existing methods like NCE, LEEP, LogME, and ModelGPT.

Learnware/Model Hubs: The paper explicitly connects to the "learnware" paradigm and the concept of model hubs. It builds upon the idea of model specification and efficient model selection in heterogeneous model repositories. MLL can be viewed as a specific instantiation of learnware principles tailored for VLMs.

**Theoretical Claims:**

There are no explicit theoretical claims made in the submission excerpt that require proof checking. The paper is primarily methodologically and empirically driven. Claims of efficiency and scalability are based on the design of the method and empirical observations, not formal theoretical derivations.

---

### Decision · Program_Chairs · 2025-05-01

**Decision:**

Accept (poster)

**Comment:**

In this paper, the authors explore how to select and reuse pre-trained VLMs for a specific downstream task, which is an important problem but has been rarely studied. To address this problem, this paper proposes a novel paradigm called Model Label Learning (MLL) that assigns each VLM a label to describe its utility on representative visual concepts, containing three key modules: model labeling,
model selection, and model reuse. The effectiveness of the proposed method is well validated by extensive experiments.

The importance of this problem is recognized by all reviewers at the beginning. However, some technical concerns still existed. After the rebuttal, all technical concerns are addressed by the authors. One remaining issue is a concern regarding the time complexity of the newly proposed framework. After checking this by the area chair and discussing it with other reviewers, this concern is minor and can be concluded as a potential limitation, instead of a reason to reject this novel paper.